# A Bioinformatics View on Acute Myeloid Leukemia Surface Molecules by Combined Bayesian and ABC Analysis

**DOI:** 10.3390/bioengineering9110642

**Published:** 2022-11-03

**Authors:** Michael C. Thrun, Elisabeth K. M. Mack, Andreas Neubauer, Torsten Haferlach, Miriam Frech, Alfred Ultsch, Cornelia Brendel

**Affiliations:** 1Department of Mathematics and Computer Science, Philipps-University Marburg, Hans-Meerwein-Straße, 35032 Marburg, Germany; 2Department of Hematology, Oncology and Immunology, Philipps-University Marburg, 35043 Marburg, Germany; 3MLL Münchner Leukämielabor GmbH, Max Lebsche Platz 31, 81377 Munich, Germany

**Keywords:** Bayesian machine learning, gene expressions, leukemia, cluster of differentiation genes, CD genes

## Abstract

“Big omics data” provoke the challenge of extracting meaningful information with clinical benefit. Here, we propose a two-step approach, an initial unsupervised inspection of the structure of the high dimensional data followed by supervised analysis of gene expression levels, to reconstruct the surface patterns on different subtypes of acute myeloid leukemia (AML). First, Bayesian methodology was used, focusing on surface molecules encoded by cluster of differentiation (CD) genes to assess whether AML is a homogeneous group or segregates into clusters. Gene expressions of 390 patient samples measured using microarray technology and 150 samples measured via RNA-Seq were compared. Beyond acute promyelocytic leukemia (APL), a well-known AML subentity, the remaining AML samples were separated into two distinct subgroups. Next, we investigated which CD molecules would best distinguish each AML subgroup against APL, and validated discriminative molecules of both datasets by searching the scientific literature. Surprisingly, a comparison of both omics analyses revealed that CD339 was the only overlapping gene differentially regulated in APL and other AML subtypes. In summary, our two-step approach for gene expression analysis revealed two previously unknown subgroup distinctions in AML based on surface molecule expression, which may guide the differentiation of subentities in a given clinical–diagnostic context.

## 1. Introduction

High-throughput molecular analyses are becoming increasingly affordable in clinical medicine, and have significantly improved our understanding of diseases such as cancer [1,2]. However, overwhelming amounts of “big omics data” still provoke the challenge of extracting meaningful information with clinical benefit. Deep molecular insights can provide a basis to shift organ/tissue-centered approaches to tumor diagnosis and therapy towards precision oncology, i.e., the integration of genetic and genomic data to estimate a patient’s prognosis and guide treatment decisions [3]. In fact, molecular classification systems have been proposed already for some entities such as breast cancer [4] or acute myeloid leukemia (AML) [5]. However, large-scale sequencing or gene expression studies are of limited practical value in specific clinical settings concerning both diagnostics and therapy. Notably, specific low-complexity biomarkers can be determined in less time than system-wide profiles, and molecular therapeutics need precisely defined structures rather than superordinate patterns as their particular sites of action. Therefore, rapid diagnostics and effective molecular therapies require knowledge about “the important few” features that low-throughput testing procedures or targeted compounds can address.

Diagnosis of acute myeloid leukemia (AML) is of great medical importance in the field of hematology, as one of its subtypes, specifically, acute promyelocytic leukemia (APL), is associated with a substantial risk of early death from bleeding complications [6]. Although diagnosis of AML can be established in many cases by cytology, discrimination of APL and non-promyelocytic AML (np-AML) is sometimes challenging and requires further examination. Throughout the past two decades, flow cytometry has become an additional powerful tool to rapidly characterize AML and distinguish this entity from other acute leukemias or myeloid neoplasms based on the immunophenotyping of blasts [7,8]. Up to now, single distinct AML-specific cell surface markers have not been identified, thus the European Leukemia Net (ELN) guidelines recommend a broad antibody panel for diagnostic testing [9].

In this work, we applied an unbiased mathematical approach that combines Bayesian and computed ABC analysis [10] on gene expression data in order to identify the most informative cluster of differentiation (CD) genes on AML blasts that could distinguish APL from np-AML. Specifically, we analyzed microarray gene expression profiles for a series of 390 primary patient samples comprising 266 np-AML, 15 APL, and 109 control cases (healthy/non-leukemia patients) [11], and RNA-Seq-data from 135 np-AML and 15 APL patient samples from the TCGA LAML project [12].

## 2. Materials and Methods

### 2.1. Gene Expression Datasets

Two AML gene expression datasets were analyzed in this study. The first one (provided by T.H) was a subset of a previously published microarray dataset [11] and included gene expression profiles of 281 diagnostic AML samples (266 np-AML, 15 APL) and 109 non-leukemia/healthy individuals that had been obtained using Affymetrix HG-U133 Plus 2.0 microarrays. The ethics board approved the analysis of this dataset for the purpose of this study at the faculty of medicine, University of Marburg (No. 138/16). The second dataset comprised RNA-Seq data from the TCGA LAML project for 135 np-AML and 15 APL patients [12], and can be downloaded from the Broad GDAC Firehose platform (http://firebrowse.org/?cohort=LAML&download_dialog=true, accessed on 3 April 2017). It can also be downloaded from the GDC Data Portal for each case separately (https://portal.gdc.cancer.gov/ (accessed on 3 April 2017). Clinical annotations for the LAML dataset included FAB subgroups as the most diverse established classification system for AML, while more recent classifications such as WHO 2016 [8] or ELN 2017/2022 classifications [7,13] were not provided. Thus, we chose FAB classes as the reference for our calculations, which we also considered appropriate given that our bioinformatic explorations were directed towards the cell surface. A list of 417 CD genes that were examined for expression in np-AML, APL, and non-leukemia samples (available only in the microarray dataset) was compiled manually using the list of CD molecules provided by the HUGO gene nomenclature committee (https://www.genenames.org/data/genegroup/#!/group/471, accessed on 30 November 2019) and a query to the NCBI gene database (https://www.ncbi.nlm.nih.gov/gene, accessed on 2 November 2017; query term “CDxxx AND Homo Sapiens [Organism]”). 

### 2.2. Calculation of Bayesian Decision Borders for Expressed and Unexpressed Genes

Initial exploratory data analysis with techniques taken from [14] indicated that the distribution of the log-transformed expression data for each of the two complete sample sets followed a multimodal distribution. Mixture models are the standard statistical tool for such applications [15,16], and further preprocessing was not required. Gaussian mixture models (GMM) were fitted to the overall distribution of the log-transformed microarray and the RNA-Seq gene expression data and subjected to Bayesian analysis using the R package “AdaptGauss” available on the Comprehensive R Archive Network (CRAN) [15]. In brief, the procedure incorporated the following definitions and operations: In Gaussian Mixture models (GMM), the probability density of the measurements, *GMM*(*x*), is represented as a weighted sum of Gaussians
(1)$$GMM\left(x\right)=\sum_{i=1}^{M}w_{i}N\left({x|\mathrm{mi}}_{i},s_{i}\right)=\sum_{i=1}^{M}w_{i}\cdot \frac{1}{\sqrt{2\pi s_{i}}}\cdot e^{-\frac{{\left(x-m_{i}\right)}^{2}}{2s_{i}^{2}}}$$
where N(x|m*_i_*,s*_i_*) denotes Gaussian normal distributions with mean m*_i_* and standard deviation s*_i_*; the weights, *w_i_*, which add up to 1, indicate the relative contribution of each component; and M denotes the number of components in the mixture. For each dataset, a suitable value for M was selected based on the Akaike information criterion [17], and the GMM was adapted to the data using an expectation–maximization algorithm. This approach resulted in a GMM with M = 2. Chi-square tests [18] were applied to estimate the probability that the GMM did not adequately describe the logarithmized gene expression data. The chi-square tests yielded *p*-values of *p* < 0.001 for both GMMs. Posteriors for each GMM were calculated as follows:(2)$$p\left(c_{i}|x\right)=\frac{p\left(c_{i}\right)*p(x|c_{i})}{\sum_{i=1}^{M}p\left(c_{i}\right)*p(x|c_{i})}$$
with
(3)$$\sum_{i=1}^{M}p\left(c_{i}\right)*p(x|c_{i})=\sum_{i=1}^{M}w_{i}*N\left(m_{i},s_{i}\right)=GMM\left(x\right)$$
in which $p(x|c_{i})$ is the likelihood to generate data in this class (conditional probability of being in the mode of gene expression values x); $p\left(c_{i}\right)$ is the probability of choosing a class (prior); and $p\left(c_{i}|x\right)$ is the posterior.

Bayesian decision borders between unexpressed and expressed genes, defined as the posteriors of p=0.5 with log expression values of x=2.96 (microarray) and x=1.48 (RNA-Seq), were calculated, for which the back transformation yielded expression values of 912 and 30.2. In detail, the left Gaussian contained low gene expression values and the right Gaussian contained high gene expression values. The posteriors defined the probability of a gene expression value either belonging to the left or to the right Gaussian. By exploiting the GMM, the expression datasets were automatically normalized to the closed interval [0, 1] using the Bayes posteriors. A value of $p\left(c_{i}|x=gene_{a}\right)=$ 0 indicates definitive underexpression of $gene_{a}$, and a value of $p\left(c_{i}|x=gene_{a}\right)=$ 1 denotes definitive overexpression of $gene_{a}$.

### 2.3. Identification of AML Subgroups

The subgroups $G_{k}$ k=1,2,3 were identified with the Ward algorithm accessible in the R package “FCPS” available on CRAN [19]. As the algorithm was applied to the Bayes posteriors $p\left(c_{i}|x\right)$, no normalization was necessary. The inspection of the dendrogram indicated optimal clustering. Here, significant changes in fusion levels of the ultrametric portion of the Euclidean distance in the Ward algorithm (y-axis) indicated the best cut [20].

### 2.4. Selection of Differentially Expressed Genes (Deg)

In the next step, the probability $\left|p_{deg}\right|$ that $gene_{a}$ is differentially expressed is defined in Equation (4) as the difference of group average Bayes posteriors $\bar{p_{G_{k}}}=\frac{1}{N_{k}}\overset{N_{k}}{\underset{u\in G_{k}}{\sum }}p_{u}\left(c_{i}|gene_{a}\right)$ with $p_{u}$ being the posterior value of a subject’s case $u\;\in G_{k}$ of a subgroup $G_{k}$ with its cardinality $N_{k}=\left|G_{k}\right|$, as follows:(4)$$p_{deg}\left(gene_{a}\right)=\bar{p_{G_{k}}}\left(c_{i}|gene_{a}\right)-\bar{p_{G_{j}}}\left(c_{i}|gene_{a}\right)$$
with $G_{k}$ and $G_{j}$ (where by k=1,2,3, j=1,2,3, j ≠k) being the two groups of comparison (e.g., APL vs. AML) and the sign (+,−) defining the direction. A value of $p_{deg}\left(gene_{a}\right)=-2$ indicates definitive underexpression, and a value of $p_{deg}\left(gene_{a}\right)=+2$ denotes definitive overexpression in relation to the compared groups. 

Differences in gene expression levels were analyzed for the dichotomies APL vs. AML1, APL vs. AML2, AML1 vs. AML2, APL vs. normal, AML1 vs. normal, and AML2 vs. normal in the microarray dataset, and APL vs. AML1, APL vs. AML2, and AML1 vs. AML2 in the RNA-Seq dataset, using Cohen’s D with varying variances and group sizes [21] as a measure of the effect size. The most important effect sizes, i.e., the most significant differences in gene expression, were identified by computed ABC analysis [10] using the R package “ABCanalysis” available on CRAN (http://cran.r-project.org/web/packages/ABCanalysis/index.html (accessed on 3 April 2017)). The ABC curve can visualize the data by graphically representing the cumulative distribution function closely related to the Lorenz curve. Using the ABC curve, the algorithm calculates the optimal limits by exploiting the mathematical properties pertaining to the distribution of analyzed items. Recursively applying ABC analysis three times on all probabilities $\left|p_{deg}\right|$ yielded the used thresholds: genes were considered to be expressed if their difference exceeded the threshold of 0.3 for RNA-Seq (0.7 for microarray data), because genes with values above the threshold were in group A (A and B respectively) in the third computed ABC analysis, and considered to be the most important ones.

## 3. Results

At first, the gene expressions in both datasets were logarithmized. Distribution analysis [14] revealed multimodal distributions for the CD genes. Therefore, distributions were modeled with Gaussian mixtures as described in the methods section. The models allowed for calculating Bayesian posteriors, which were used for further analysis.

### 3.1. Subgroups of AML Based on the Expression of CD Genes

In the next step, hierarchical clustering, using the Ward algorithm, of the RNA-Seq data revealed three major clusters marked in the dendrogram in black in Figure 1, consisting solely of APL/AML M3 cases (Ward cluster 3). Beyond this separation, AML had two subgroups (magenta and red). In order to compare the RNA-expression-based AML clustering with the historical classification of the French–American–British group [22], both classifications were aligned in Table 1. Within the first cluster of Ward, M1, M2, and M0 were predominant, and within the second cluster, M5. M4 was equally distributed, and for M6 to M8, only a few cases were available. FAB M3 is historically defined as APL, and was only found in Ward cluster 3. Hence, the Ward dendrogram for the AML patients without APL of the microarray dataset was also investigated. In line with the clustering results from the RNA-Seq data, the non-APL group revealed a cluster structure of two subgroups, as depicted in Figure 2.

A dichotomy of np-AML is neither recognized in the present WHO classification [7] nor in the stratification for treatment strategies. The separation of np-AML to APL is in agreement with previous research [23]. In addition, it is mentioned in one prior publication that AML divides into one APL cluster and two AML clusters with principal component analysis using 37 microarray samples [24]. Alignment of the two clusters to FAB classification could either be interpreted as a myeloid vs. monocytic differentiation pattern or as an immature (M0–2 and M6/M7) vs. a more mature (M4–M5) biologic nature of the leukemia blasts, as suggested by Goardon et al. [25]. To corroborate the latter hypothesis, we performed a detailed inspection of specific antigen distribution patterns. 

### 3.2. Differential Expression of CD Genes in AML Subtypes

To determine the CD genes that most accurately discriminate the subgroups of AML (APL, AML1, AML2) in the microarray analysis, we determined the differences in mean expression values for each dichotomy by calculating Cohen’s D and then performed computed ABC analyses. In total, 23 genes significantly discriminated two AML subgroups in at least one dichotomy in the microarray dataset (Table 2). Moreover, 10 genes were differentially expressed in APL vs. AML1, and 19 genes in APL vs. AML2, with six co-regulated genes in APL vs. both groups. Only one gene (CD18) was differentially expressed in all three pairs of AML subgroups. Hence, a comparison of APL vs. a mixture of both AML groups would have underscored the expression of considerable many potential target genes.

Of six genes that were correspondingly differentially expressed between APL and each AML1 and AML2, three (CD3D, CD98, CD339) were higher expressed in APL, and three (CD52, CD62L, CD83) were higher in AML1/2. In addition, the CD18 gene was also higher in APL vs. AML1/2, and differentially expressed in all three pairs of AML subgroups. Finally, one gene (CD354) discriminated AML1 from AML2, but not AML1 and AML2 from APL. 

In the RNA-Seq dataset, 14 genes significantly discriminated between AML subgroups in at least one dichotomy (Table 3). Three genes (CD84, CD148, CD339) separated APL from each AML1 and AML2, with all of these genes showing higher expression in APL. In addition, three genes (CD91, CD197, CD261) discriminated between AML1 and AML2. However, CD197 did not discriminate in the microarray data. Of note, CD84, CD148, CD197, and CD261, as well as all other genes that significantly discriminated between two AML subgroups only in one dichotomy (APL vs. AML1, APL vs. AML2, AML1 vs. AML2), were only found in one dataset and showed a Cohen’s D-value of 0 in the other and CD91 was not expressed in the microarray dataset (Appendix A). Thus, overexpression of CD339 in APL was the only consistent finding in the two datasets.

Together, the comparison of APL vs. AML2 revealed a diverse expression of well-known differentiation marker molecules, such as CD11b, CD13, CD14, CD15, and CD24, whereas fewer surface molecules were differentially expressed between APL and AML1. The minor expression of specialized functional molecules inside the cell and on the surface is a feature of “stemness”, hereby indicating that AML2 might represent a more differentiated type of AML, and that AML1 is more stem-cell-like. Moreover, only CD79a was overexpressed in AML1 vs. APL. CD79a is considered a B-cell antigen, but is often expressed in immature blast crisis acute leukemias following CML (chronic myeloid leukemia) [26].

### 3.3. Differential Expression of CD Genes in AML Compared to Normal Samples

To further substantiate the differential expression of specific CD genes in AML1, AML2, and APL, we compared these entities to normal samples separately. The results are presented in Table 4. As TCGA RNA-Seq data did not include healthy/normal controls, these analyses were performed only in the microarray dataset. ABC analyses revealed differential regulation between at least one AML subgroup and normal controls of 48 CD genes. 

Two genes (CD99, CD135) were upregulated in each AML subgroup compared to healthy/normal samples, and five genes (CD24, CD66b, CD66c, CD218b, CD233) were expressed significantly lower. One gene (CD88) showed opposing differential regulation in APL and AML1 compared to AML2 relative to healthy bone marrow. Three genes were differentially expressed only between APL and normal, with two genes (CD3D, CD339) being upregulated and one gene (CD50) being downregulated. Furthermore, 20 genes were expressed higher and 6 genes lower in AML2, but no gene was discriminative for AML1 alone or AML1/2 compared to normal (Table 3). Four genes (CD99, CD135, CD218b, and CD233) separated any of the AML subgroups from normal bone marrow without discriminating between them. On the other hand, 21 genes (CD3D, CD9, CD11b, CD14, CD18, CD24, CD37, CD50, CD52, CD62L, CD66b, CD66c, CD83, CD87, CD88, CD98, CD210A, CD282, CD339, CD354, and CD369) were differentially expressed between both at least two different AML subgroups and one AML subgroup compared to normal samples (Table 2 and Table 3). Thus, ABC analysis using normal samples as a reference confirmed the differential expression of CD genes in different subgroups of AML.

### 3.4. Validation of Target Genes by Literature Screening

In order to corroborate the target CD antigens with a focus on the distinction between APL and np-AML that were derived from both microarray and RNA-Seq datasets, we screened the literature for reports from other AML groups [Table 5].

In sum, 36 genes (14 in the RNA-Seq, 23 in the microarray dataset) were expressed differentially in APL and np-AML (Table 5). Overexpression of CD339/JAG1 in APL was the only consistent finding in the two datasets, in line with previous reports including both gene expression data from RNA-Seq and microarray analyses [27] or flow cytometry studies [28] Of note, differential expression in APL and other AML subtypes cannot be deduced from these earlier studies for all CD genes identified by our calculations, such as CD84 [29], CD88 [30], CD148 [23,31], CD210A/IL10RA, and CD227 [32,33]. However, all of these genes have been detected in cells from all hematopoietic lineages or AML blasts, and CD148 and CD210 have even been recently suggested as novel immunotherapeutic targets in AML [34], except forCD261/TNFRSF10A which we did not find in previous publications pointing to implications in AML or APL. 

Several genes that discriminated AML or one of its major subgroups from controls are known diagnostic markers and/or targets for investigational therapies in AML, for example CD11b, CD13, CD14, CD24, CD33 [35,36], CD71 [37], CD81 [38] and CD99 [39], while CD132 has not been investigated concerning its potential role in AML.

Therefore, current knowledge of CD gene expression in distinct subgroups of AML strongly supports the validity of our mathematical approach to extract clinically important information from gene expression data, specifically in the context of AML, for diagnostic purposes.

**Table 5 bioengineering-09-00642-t005:** Supporting evidence from the literature for differential expression of CD genes in APL and other AML subtypes. If differential expression could not be deduced from previous reports, references describing expression in AML are indicated.

Index in Table 2/Table 3	Gene	Evidence
21,14	CD339	[27,28]
1	CD3D	[40]
2	CD9	[41,42,43,44]
3	CD11b	[41,42,43,44]
4	CD14	[19,43]
5	CD15	[45,46]
6	CD18	[47]
7	CD24	[48]
8	CD37	[49]
9	CD50	[50]
10	CD52	[51]
11	CD55	[52]
12	CD62L	[53]
13	CD66b	[54]
14	CD66c	[54]
15	CD83	[55,56]
16	CD87	[57]
17	CD88	[30]
18	CD98	[58,59]
19	CD210A	[60,61]
20	CD282	[57]
23	CD369	[62]
1	CD1D	[63]
2	CD7	[64,65]
3	CD13	[45,46]
4	CD44	[60,66]
5	CD79A	[67,68]
6	CD84	[29,69]
7	CD88	[30]
9	CD148	[31]
11	CD227	[24,32,33]
12	CD230	[70]
13	CD261	x

## 4. Discussion

Characteristic expression patterns of cell surface molecules provide the basis for rapid diagnosis of AML by flow cytometry [9]. Here, we used combined Bayesian and ABC analysis to identify subgroups of AML based on characteristic cell surface patterns reconstructed from gene expression data for CD genes. We explored associations of AML CD subgroups with an established morphologic classification system for AML, precisely, FAB classification. Moreover, we extracted the most important discriminating CD genes that distinguish the AML subgroups from each other and normal samples. As the focus of our work was more closely related to potential diagnostic applications, the following discussion primarily refers to current experimentally confirmed knowledge on CD gene expression in AML. Yet, clinical implications are clearly emerging as a future perspective, given the growing spectrum of monoclonal antibodies targeting AML and related myeloid diseases [71].

Intriguingly, using two independent AML expression datasets obtained by microarray or RNA-Seq analyses, we found that AML separated into only two natural clusters in both datasets based on the expression of CD genes. In the RNA-Seq dataset, we identified two subgroups of AML and were able to separate APL from AML. One of these clusters is APL, the only subclass of AML that has been previously defined, and is managed therapeutically differently from all other AML subclasses in the clinic [72]. Np-AML, on the other hand, is less diverse based on CD gene expression than, for example, based on morphology or genetic findings. Employing the Ward algorithm on 394 CD genes, AML clustered into two subgroups, which we here pragmatically designated AML1 and AML 2. In the RNA-Seq dataset, AML1 was associated with FAB classes M1 and M2, while AML2 corresponded to FAB class M5. Both AML subgroups contained FAB class M4. In the microarray dataset, the FAB classification was not accessible. This association supports the notion that the AML1 cluster refers to an immature immunophenotype, while the AML2 cluster characterizes leukemia cells with the expression of maturation markers, i.e., AML1 cells may derive from a pluripotent stem cell and AML2 from a progenitor cell, respectively.

In order to further investigate whether the classification of AML by CD gene expression may be exploited for AML diagnosis by flow cytometry in addition to previously recommended immunophenotype targets, we selected the most important discriminating genes for each subgroup compared to the other subgroups and to normal controls. Performing ABC analysis on the microarray and the RNA-Seq datasets predominantly yielded non-overlapping results. Only the CD339 gene, which corresponds to the Jagged-1 protein, was expressed higher in APL than in AML1 and AML2 in both datasets, and was also distinctive for APL compared to normal samples, which is in line with previous analyses of publicly available gene expression datasets [27]. However, CD339 is not included in previously reported antibody panels for the identification of APL by flow cytometry [41,42,73]. Therefore, it is tempting to speculate that our unbiased analysis of the AML surface can inform the design of improved diagnostic tools. 

On the other hand, the only other gene that was somehow discriminating between AML subgroups, CD88, was underexpressed in AML2 compared to APL in the RNA-Seq dataset but overexpressed relative to both APL and AML1 in the microarray dataset. CD88 (C5AR1) has been reported to be expressed in AML cell lines and primary patient blasts and to contribute to cell motility [30], illustrating its potential role in this disease. We attribute the conflicting results for CD88 expression, as well as the overall lack of consistent observations, to the different molecular techniques for gene expression measurement [74], as Bayesian analysis based on GMMs has been shown to yield stable and consistent results on noisy data of various types, such as income, heat or cold pain, GDP time series or nitrate concentrations in streams [15,75,76]. Of note, the dataset of only 15 cases of APL samples is small, but in line with the reported frequency of this subgroup. Consequently, consolidation with a larger dataset would be desirable to strengthen the results reported here. Of non-overlapping genes that were differentially expressed between AML subgroups or AML and normal samples, several CDs are either known targets (such as CD33) or have already been reported to be implicated in AML.

Novel treatment options for AML cases with an unfavorable prognosis are highly demanded and monoclonal antibodies that target AML surface molecules are currently evaluated in clinical trials ([71,77]). Although we do not have detailed information on cytogenetics or survival in both of our datasets, our study reveals potential target molecules, particularly if AML1 is indeed a stem cell near a subgroup with inferior prognosis.

In summary, we conclude that our unbiased mathematical approach was highly successful in deducing knowledge from complex biological datasets. “The important few” AML-specific CD genes identified by combined Bayesian and calculated ABC analysis are widely congruent with actual knowledge in the field, independent from the particular method of gene expression analysis. Thus, the work described here may serve as a blueprint for translating relevant information from high-dimensional biological data into clinical applications.

## Figures and Tables

**Figure 1 bioengineering-09-00642-f001:**
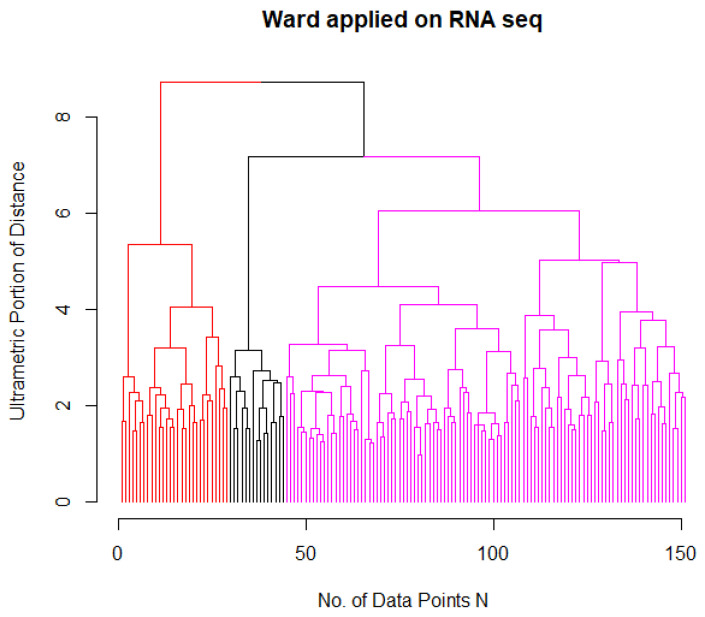
Ward clustering applied to the RNA-Seq data revealed a cluster structure of three groups $G_{k}$, k=1,2,3, one consisting only of APL and two of AML with the FAB classification of Table 1 which also depicts the colors of this figure. Clustering and dendrogram generation were computed with the R package “FCPS” available on the Comprehensive R Archive Network (CRAN) [19].

**Figure 2 bioengineering-09-00642-f002:**
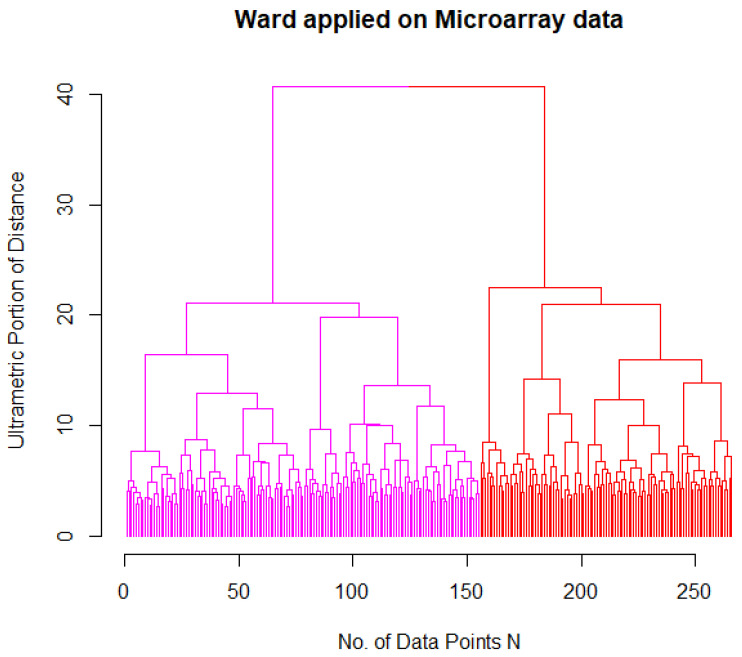
Dendrogram of Ward clustering applied to microarray np-AML patient samples revealed two subgroups (magenta and red). Clustering and dendrogram generation were computed with the R package “FCPS”, available on CRAN. The dataset provided normal controls in addition to APL. Both diagnostic entities were not used in the clustering.

**Table 1 bioengineering-09-00642-t001:** Contingency table comparing Ward clustering of the RNA-Seq dataset and FAB classification [22]. Abbreviations: M0, undifferentiated acute myeloblastic leukemia; M1, acute myeloblastic leukemia with minimal maturation; M2, acute myeloblastic leukemia with maturation; M3, acute promyelocytic leukemia (APL); M4, acute myelomonocytic leukemia; M5, acute monocytic leukemia; M6, acute erythroid leukemia; M7, acute megakaryoblastic leukemia.

Ward Cluster	M0	M1	M2	M3	M4	M5	M6	M7
1 (magenta)	15	34	36	0	17	2	2	1
2 (red)	0	3	1	0	12	13	0	0
3 (black)	0	0	0	15	0	0	0	0

**Table 2 bioengineering-09-00642-t002:** The most important differentially expressed CD genes in AML subtypes in the microarray dataset. Cohen’s D values are indicated for each dichotomous comparison (group A on the left vs. group B on the right). Positive values indicate higher expression in group A. Negative values indicate higher expression in group B. Values < 0.7 were set to 0.

	Gene	APL vs. AML1	APL vs. AML2	AML1 vs. AML2
1	CD3D	0.9	0.92	0
2	CD9	0.89	0	0
3	CD11b	0	−1.17	−0.94
4	CD14	0	−0.99	0
5	CD15	0	−0.77	0
6	CD18	−0.97	−1.69	−0.72
7	CD24	0	−1.29	−0.79
8	CD37	0	−1.09	0
9	CD50	0	−0.98	0
10	CD52	−0.96	−1.25	0
11	CD55	0.78	0	0
12	CD62L	−1.04	−1.44	0
13	CD66b	0	−1.49	−0.82
14	CD66c	0	−1.06	0
15	CD83	−0.7	−1.4	0
16	CD87	1.08	0	0
17	CD88	0	−1.1	−1.09
18	CD98	0.93	1.05	0
19	CD210A	0	−1	−0.79
20	CD282	0	−0.81	−0.75
21	CD339	0.9	0.75	0
22	CD354	0	0	−0.75
23	CD369	0	−1.04	−0.94

**Table 3 bioengineering-09-00642-t003:** The most important differentially expressed CD genes in AML subtypes in the RNA-Seq dataset. Cohen’s D values are indicated for each dichotomous comparison (group A vs. group B). Positive values indicate higher expression in group A. Negative values indicate higher expression in group B. Values < 0.3 were set to 0.

	Gene	APL vs. AML1	APL vs. AML2	AML1 vs. AML2
1	CD1D	0	0.42	0
2	CD7	0	0.52	0
3	CD13	0	0.39	0
4	CD44	0	0.53	0
5	CD79A	−0.39	0	0
6	CD84	0.32	0.32	0
7	CD88	0	0.34	0
8	CD91	0	0	−0.32
9	CD148	0.88	0.89	0
10	CD197	0	0	0.32
11	CD227	0	−0.36	0
12	CD230	0	0.32	0
13	CD261	0	0.55	0.52
14	CD339	0.45	0.42	0

**Table 4 bioengineering-09-00642-t004:** The most important differentially expressed CD genes between AML subtypes and normal controls in the microarray dataset. Cohen’s D values are indicated for each dichotomous comparison (group A/left vs. group B/right). Positive values indicate higher expression in group A. Negative values indicate higher expression in group B. Values < 0.7 were set to 0.

	Gene	APL vs. Normal	AML1 vs. Normal	AML2 vs. Normal
1	CD1D	0	0	0.39
2	CD3D	0.72	0	0
3	CD9	1	0	0.31
4	CD11b	−1.24	−1	0
5	CD13	0	0	0.33
6	CD14	0	0	0.49
7	CD18	−1.18	0	0.52
8	CD24	−1.83	−1.33	−0.54
9	CD32	0	0	0.34
10	CD33	0	0	0.46
11	CD37	0	0	0.97
12	CD44	0	0	0.45
13	CD50	−0.73	0	0
14	CD52	−0.86	0	0.39
15	CD62L	−0.98	0	0.46
16	CD66b	−1.91	−1.24	−0.42
17	CD66c	−1.8	−1.43	−0.74
18	CD71	0	0	−0.44
19	CD81	0.82	0	0.53
20	CD83	−0.84	0	0.56
21	CD87	0.8	0	0.4
22	CD88	−0.8	−0.79	0.3
23	CD97	0	0	0.73
24	CD98	1.01	0	0
25	CD99	1.44	1.14	1
26	CD107b	0	0	0.41
27	CD114	0	0	0.55
28	CD119	0	0	0.39
29	CD120b	0	0	0.5
30	CD132	0.72	0	0.35
31	CD135	0.7	0.82	0.6
32	CD157	0	0	0.37
33	CD210A	0	0	0.68
34	CD218b	−0.79	−0.7	−0.52
35	CD225	0	0	−0.39
36	CD230	0	0	0.36
37	CD233	−1.29	−0.83	−1.09
38	CD235a	−0.87	0	−0.81
39	CD235b	0	0	−0.53
40	CD236	0	0	−0.41
41	CD240D	0	0	−0.44
42	CD241	0	0	−0.31
43	CD256	0	0	0.4
44	CD257	0	0	0.53
45	CD282	0	0	0.68
46	CD339	0.92	0	0
47	CD354	0	0	0.75
48	CD369	0	0	0.4

## Data Availability

RNA-Seq data can be downloaded from the GDC Data Portal (https://portal.gdc.cancer.gov/ (accessed on 3 April 2017). In addition, the RNA-seq data can be provided upon request. (accessed on 3 April 2017)). The microarray dataset was published by Haferlach et al. [11].

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
