# Peer review of "A Bioinformatics View on Acute Myeloid Leukemia Surface Molecules by Combined Bayesian and ABC Analysis"

_bioengineering, 2022, doi:10.3390/bioengineering9110642_

Round 1

Reviewer 1 Report

The report by Thrun et al provides a highly innovative bioinformatic tool for the assessment of surface molecules on blasts of acute myeloid leukemia (AML). The scientific question is highly relevant, the study is rigorously designed, the data appear to be robust and the conclusions drawn by the authors are supported by the experimental evidence that has been collected. 

My only suggestion is to include a sentence on the potential relevance of this bioinformatic analysis for informing treatment choices with the many monoclonal antibodies targeting AML surface molecules, as covered in detail in a recent review that should be cited (Gallazzi et al., New Frontiers in Monoclonal Antibodies for the Targeted Therapy of Acute Myeloid Leukemia and Myelodysplastic Syndromes. Int J Mol Sci. 2022 Jul 7;23(14):7542. doi: 10.3390/ijms23147542). 

Author Response

We thank reviewer 1 for emphasizing the therapy related aspect of our study on AML surface molecules. We now mentioned antibody-based AML therapy in the the discsussion (line 348-350) and cited the recommended publication In addition we elaborated the focus of the work in follows in Lin 299ff:
As the focus of our work was more closely related to potential diagnostic applications, the following discussion primarily refers to current experimentally confirmed knowledge on CD gene expression in AML. Yet, clinical implications are clearly emerging as a future perspective, given the growing spectrum of monoclonal antibodies targeting AML and related myeloid diseases(1).

Reviewer 2 Report

Manuscript „A Bioinformatics View on Acute Myeloid Leukaemia Surface Molecules by Combined Bayesian and ABC Analysis” provides possible approach for use of big data analysis in diagnostics. It shows how mathematical approach can help distinguish relevant biological information and yield the results that can be confirmed by wet-lab research.

However, there are two points that should be additionally explained and discussed:

1)    Used information include only 15 cases of APL compared to more than 200/100 AML/non-tumor bone marrow cases which is a very small number of cases for reliable comparison

2)    Why was FAB classification used instead of current WHO classification?

Author Response

  1. Used information include only 15 cases of APL compared to more than 200/100 AML/non-tumor bone marrow cases which is a very small number of cases for reliable comparison.

Among all AML subgroups APL (AML M3) constitutes 5-10% - in line with the frequency reported in our data sets. We totally agree with the reviewers comment that particularly from the mathematic view a confirmation of our calculations with a bigger patient cohort would be desirable. We therefore added a comment in the discussion (line 343ff).

2. Why was FAB classification used instead of current WHO classification?

 We agree with the reviewer that the FAB classification of AML is only of minor importance in the current clinical management of AML. We added the following sentence to Materials and Methods in order to explain why our analyses used FAB soubgroups as an example for an established AML classification system to which results were related in Line 76ff:
Clinical annotations for the LAML dataset included FAB-subgroups as the most diverse established classification system for AML, while more recent classifications such as WHO 2016 [8] or ELN 2017/2022 classifications [22],(2) were not provided. Thus, we chose FAB classes as the reference for our calculations, which we also considered appropriate given that our bioinformatic explorations were directed against the cell surface.

Reviewer 3 Report

1. Based on the nature of the study, the results are compared with the FAB classification, which is old, still valid in several cases. Could there be a comparison and/or application of the results regarding the ELN 2022 (or ELN 2017) classification of AML? PLease comment on the discussion. 

2. The diagnosis of APL can be made in the clinic by the combination of clinical, laboratory data and/or immunophenotype. How does this study help in diagnosing APL?

3. Could you make a list of known immature vs more mature leukemias, like the comparison of AML1 vs AML2 cells?

4. Is your analysis restricted in diagnosing AML or are there applications for the prognosis of AML also? Does this study add to the diagnosis of AML or is it just mathematics confusing the clinicians?

5. In general, the results are complicated. Please condense them. 

Overall, it is an interesting and well conducted study.  

Author Response

attached. Our responses in red.
